# The Role of Honey Bee Derived Aliphatic Esters in the Host-Finding Behavior of *Varroa destructor*

**DOI:** 10.3390/insects14010024

**Published:** 2022-12-25

**Authors:** Jiamei Liu, Ruonan Zhang, Rui Tang, Yi Zhang, Rui Guo, Guojun Xu, Dafu Chen, Zachary Y. Huang, Yanping Chen, Richou Han, Wenfeng Li

**Affiliations:** 1College of Animal Sciences (College of Bee Science), Fujian Agriculture and Forestry University, Fuzhou 350002, China; 2Guangdong Key Laboratory of Animal Conservation and Resource Utilization, Guangdong Public Laboratory of Wild Animal Conservation and Utilization, Institute of Zoology, Guangdong Academy of Sciences, Guangzhou 510260, China; 3School of Chinese Medicinal Resource, Guangdong Pharmaceutical University, Yunfu 527527, China; 4Department of Entomology, Michigan State University, East Lansing, MI 48824, USA; 5USDA-ARS Bee Research Laboratory, Beltsville, MD 20705, USA

**Keywords:** honey bee, *Varroa destructor*, host finding, aliphatic esters

## Abstract

**Simple Summary:**

Honey bees provide essential pollination services for many agricultural crops and wild plants. However, honey bee health is at risk. As an obligate ectoparasite of honey bees, *Varroa destructor* has been causing huge damage to the worldwide Western honey bee colonies since the host shift from its original host, the Eastern honey bees. For years, a lot of effort has been made to control this parasite; however, more efficient and safer methods are still urgently needed. The current study also aimed to contribute to the *Varroa* mitigation by seeking the potential compounds that affected the host-finding behavior of *Varroa* mites. By profiling the headspace volatiles of honey bee larvae, four aliphatic esters were commonly identified from both worker and drone larvae of the original and new hosts. Among the esters, ethyl myristate was a novel compound and able to significantly attract the *Varroa* mites. The findings presented in this paper will deepen our understanding of the *Varroa* host finding behavior and provide new chemicals for the monitoring and control of *V. destructor* in the future.

**Abstract:**

*Varroa destructor* is an obligate ectoparasite of honey bees and shifted from its original host *Apis cerana* to the new host *Apis mellifera* in the first half of the twentieth century. The host shift has resulted in a great threat to the health and survival of *A. mellifera* colonies worldwide. Chemical signals play a crucial role in all aspects of the *Varroa* life cycle, including host finding. However, the chemical cues that affect the host finding behavior of *Varroa* mites are still not fully understood. In this study, we systematically profiled the headspace volatiles of both worker and drone larvae of the two honey bee species by using solid phase micro-extraction coupled to gas chromatography-mass spectrometry (SPME-GC-MS), and then used electrophysiological recording and Y-tube olfactometer bioassay to study the potential roles of the selected compounds. The chemical profiling showed that there were four aliphatic esters, ethyl myristate (EM), methyl palmitate (MP), ethyl palmitate (EP), and ethyl oleate (EO) commonly detected from all four types of larval hosts. Among them, EM was a new substance identified from honey bee headspace volatiles. Results from electrophysiological recordings indicated that all the aliphatic esters could elicit significant responses of *Varroa* pit organs on its forelegs. Moreover, behavioral analyses revealed that EM could significantly attract *V. destructor* at a medium dosage (10 µg), while MP had no observable effect on the mites and both EP and EO were able to repel the parasites. Our findings suggest an important role of host-derived aliphatic esters in *Varroa* host finding, and provide new chemicals for *Varroa* monitoring and control.

## 1. Introduction

As important economic insects, honey bees (genus: *Apis*) not only provide multiple bee products for human well-being, but also pollinate many agricultural crops and wild flora. Both bee pollination and bee product production depend on healthy colonies. However, in recent years, large-scale colony losses have been reported in several parts of the world, suggesting that the health of honey bee colonies has been under unprecedented threat [1,2,3,4]. Currently, it is known that many factors have caused the high mortality of bee colonies, notably the invasion and outbreak of emerging parasites and pathogens [5,6,7,8].

Among the parasites that negatively impact bee health, *Varroa destructor* is thought to be the most harmful to the Western honey bee, *Apis mellifera* [9,10,11]. In order to control this deadly parasite, many methods have been developed, and the most commonly used means are several synthetic acaricides such as organophosphorus coumaphos (Checkmite, Asuntol, Perizin), pyrethroids (Apistan, Klartan, Mavrik), flumethrin (Bayvarol), and amitraz (Apivar) [11,12]. Although these chemicals are effective to a certain extent, pesticide resistance has rapidly developed in *Varroa* populations [13,14,15,16]. Additionally, as soft acaricides organic acids and essential oils are also commonly used to control *Varroa* mites, however, there are some shortcomings reported when using these natural compounds [11]. Therefore, safer, more efficient and more reliable means of preventing and controlling *V. destructor* are still urgently needed.

*V. destructor* is an ectoparasitic mite originally found in the eastern honey bee *A. cerana*. Since its successful shift to *A. mellifera* in the first half of the twentieth century, *V. destructor* has rapidly expanded to most *A. mellifera* colonies worldwide [11,17]. There is a close link between the entire life cycle of the *Varroa* mites and the honey bee hosts. The life cycle of *V. destructor* is divided into two phases: the phoretic phase, during which the mites attach themselves to adult bees and feed on their fat bodies [18], and the reproductive phase, mainly completed inside the brood cells [11,17].

Volatile organic compounds (VOCs) play crucial roles in interspecific and intraspecific communications, and numerous studies suggest VOCs as communication signals [19]. For instance, VOC cues emitted by neighboring guava plants could trigger defense responses in sweet orange (*Citrus sinensis*) by boosting jasmonate-dependent anti-herbivore activities [20]. Foliar application of *Sophora alopecuroides* alkaloid extract affected the Asian citrus psyllid *Diaphorina citripsyllid* host-finding behavior [21]. Additionally, VOC emissions from food sources influenced intra- and interspecific interactions among stored-product Coleoptera in paddy rice [22]. Host finding is one of the key activities during the life cycle of *V. destructor*, which further affects its reproduction and transmission. Factors stimulating the host finding of *V. destructor* might be complex and diverse, however, chemical stimuli are found to be essential and play a fundamental role in *Varroa* host finding [11,17,23]. Several fractions of the cuticular extract from honey bee larvae have been found to be attractive to *Varroa* mites. Le Conte et al. (1989) [24] first reported that three simple aliphatic esters, methyl palmitate, ethyl palmitate, and methyl linolenate from *A. mellifera* larval cuticle triggered significant attraction to *V. destructor*, and drone larvae contained higher levels of these compounds than worker larvae, supporting the preferred invasion of drone brood [25,26,27]. On the other hand, as components of honey bee brood pheromone, these esters can elicit capping behavior in hive bees [28], which suggests they act as both pheromone and kairomone in honey bee colonies. However, only the pheromonal function was confirmed by a subsequent study, where paraffin wax dummies with the esters applied induced capping behavior in the hive bees, but had no effect on *V. destructor* [29]. By using a servosphere, Rickli et al. (1992) [30] demonstrated that palmitic acid identified from the headspace extract of worker larvae could significantly attract *Varroa* mites, while methyl palmitate evoked a weak response. 

Insect cuticular hydrocarbons (CHCs) are compounds that usually serve as chemical cues mediating various activities like nestmate recognition, chemical mimicry, and camouflage [31]. *Varroa* mites were found arrested strongly by simple odd numbered C19–C29 *n*-alkanes, the major constituents of the cuticular extract of newly-capped worker larvae [32]. Moreover, the fourth and fifth instar larvae had different CHC profiles and distinct attractiveness to *Varroa* mites, indicating that the parasites may determine the host age by sensing CHCs [33]. More recently, Li et al. (2022) [34] revealed that the larval CHC profiles between the original host (*A. cerana*) and new host (*A. mellifera*) were significantly different, which gave rise to the host preference of *V. destructor*. *Varroa* mites can be attracted by not only host-derived compounds, but also the semiochemicals from brood-related sources such as cocoon and larval food. On the other hand, there are also chemicals inside the honey bee hives that show a repellent effect on the parasites [11,17]. In brief, chemical stimuli that affect the host finding behavior of *Varroa* mites are complex and diverse, and despite the known compounds it is likely to identify novel chemicals that may have stronger attractiveness/repellences to *V. destructor*. 

In the current study, we assume that there are still some new compounds that can be derived from honey bee hosts and affect *Varroa* host finding behavior. We used a more recently-developed method to enrich and analyze the headspace volatiles from the worker and drone larvae of both the original and new host species. Since both drone and worker larvae in both species of honey bees can effectively attract *Varroa* mites [34,35], we compared and selected the common volatiles and then performed several electrophysiological analyses and Y-tube olfactometer bioassays. The forelegs of *V. destructor* were used to record the electrophysiological responses as the pit organ located on the foreleg tarsi was found to sense the headspace volatiles of adult honey bees [36,37]. We found out one novel and three known compounds, and revealed their attractive/repellent effect on *Varroa* mites during host finding.

## 2. Materials and Methods

### 2.1. Biological Samples

Both *A. mellifera* (Am) and *A. cerana* (Ac) colonies used in this study were kept in the same experimental apiary of the Institute of Zoology, Guangdong Academy of Sciences, Guangzhou, China. The *A. mellifera* colonies were further classified and labeled as *Varroa*-free or *Varroa*-infested colonies. Early fifth instar larvae were sampled by uncapping the sealed brood cells and collecting those with bowed backs and not spinning silk yet. Both worker and drone larvae were obtained from *Varroa*-free *A. mellifera* and *A. cerana* colonies.

The female adults of *V destructor* were taken from sealed brood cells in heavily-infested *A. mellifera* colonies. All *Varroa* mites were maintained on the newly emerged bees for 48 h before physiological or behavioral tests. 

### 2.2. Chemicals

Ethyl palmitate (>99.0% purity) was purchased from Dr. Ehrenstorfer GmbH (Augsburg, Germany). Ethyl oleate (>99.0% purity), methyl palmitate (>99.0% purity) and ethyl myristate (>99.0% purity) were purchased from ANPEL (Shanghai, China). 1-nonene (>99.0% purity) was purchased from Sigma-Aldrich (St. Louis, MO, USA). Ethanol (HPLC-grade) and hexane (HPLC-grade) were purchased from Kermel (Tianjin, China).

### 2.3. Chemical Analysis

Headspace volatiles on the body surface of honey bee larvae were sampled using solid phase micro-extraction (SPME), and analyzed by gas chromatography-mass spectrometry (GC-MS). Briefly, for the SPME sampling, individual honey bee larvae sampled as described above were sealed in a 20 mL glass vial and 1 μL of 1-nonene ethanol solution (1:10,000, *v*/*v*) was also added to the vial as an internal standard. The headspace extraction was performed with a 50/30 μm DVB/CAR/PDMS stableflex SPME fiber (Supelco, Bellefonte, PA, USA) at 34 °C for 6 h. The enriched volatile compounds were then subjected to GC-MS analysis on an Agilent 7890B GC system (equipped with an HP-5MS UI column; film thickness: 0.25 µm; length: 30 m; inner diameter: 0.250 mm) connected to an Agilent 5977 MSD (quadrupole mass spectrometer with 70-eV electron impact ionization). The back inlet was at 260 °C and 7.06991 psi, and the carrier gas helium was provided at a constant flow rate of 34 mL/min. Samples were added in splitless mode. Oven temperature was programmed at the following temperatures: 40 °C to 48 °C at 4 °C/min, 48 °C to 60 °C at 1.5 °C/min, 60 °C to 82 °C at 4 °C/min, 82 °C to 100 °C at 1.5 °C/min, 100 °C to 200 °C at 4 °C/min, and 200 °C to 260 °C at 6 °C/min. The mass spectrometer was set to scan from 33–390 *m*/*z*. The MS source and quadrupole were maintained at 230 °C and 150 °C, respectively. GC-MS data were analyzed using MSD ChemStation (G1701FA F. 01. 03. 2357) in MassHunter software suites (Agilent Technologies, Palo Alto, CA, USA). Compounds were identified by searching spectra against the NIST mass spectral libraries (NIST17) in MassHunter. The peak area of each compound was obtained by automatic integration of each total ion chromatogram (TIC) using a consistent baseline setting. The concentration of every compound was calculated by comparing with the peak area of the known amount of internal standard 1-nonene [38].

### 2.4. Electrotarsogram (ETG) Recordings

ETG recordings were performed on *V. destructor* foreleg exposed to test chemicals at room temperature, according to methods described in previous studies with modifications [36,39]. Briefly, the foreleg was excised at the base under a Zeiss Discovery V8 dissection microscope (Zeiss, Jena, Germany) and mounted to each side of a two-pronged electrode “fork” recording probe with a small amount of electrode gel (Spectra 360, Parker Laboratories, Fairfield, NJ, USA). A humidified and charcoal-filtered air flow provided by a stimulus controller (Syntech CS-55, Hilversum, The Netherlands) was constantly blown towards the organ at 30 mL/s at a distance of 1 cm. 

To prepare odor cartridges, 10 µL of test or control solution was added to a piece of Whatman No. 1 filter paper (1 cm × 1 cm) which was placed in a glass Pasteur pipette and exposed to air for 30 s for evaporation of the solvent. Chemicals were dissolved in hexane to prepare the test solutions (100 mg/mL). In order to assess dosage response, a serial dilution was carried out to generate different concentrations of stimuli (0.01, 0.1, 1, 10 and 100 mg/mL). Therefore, dosages of each stimulation were 0.1, 1, 10, 100, and 1000 µg. 

Each odor was puffed at the foreleg for 200 ms, followed by a two-minute interval to allow recovery of the organ. The resulting response was amplified and recorded by a signal acquisition interface board (IDAC-2, Syntech, Hilversum, The Netherlands), and then the recordings were analyzed by AutoSpike software (Syntech, Irvine, CA, USA). The olfactory response for each test stimuli was calibrated using the mean value of the two responses of control (solvent alone).

### 2.5. Y-Tube Olfactometer Bioassays

A Y-tube olfactometer bioassay was designed to test the behavioral response of *V. destructor* to the volatiles derived from honey bee larvae, modified from Pernal et al. (2005) [40]. The main part of the olfactometer was a glass Y-tube (stem and arms: length 5 cm, diameter 2 cm). A marker was made on the stem of the Y-tube, 3 cm from the junction of the ‘‘Y’’, where a mite would be introduced. Each arm of the Y-tube was connected to a cylindrical glass odor chamber. The anterior end of each odor chamber could be removed and replaced, thus allowing sample insertion and removal. Ambient air was charcoal-filtered, moistened with distilled water, and then blown into the two arms using a vacuum pump. The velocity of the airflow passing through the glass odor chambers was adjusted to 100 mL/min using an airflow meter (Kean Labor Insurance, Beijing, China). The experimental condition was kept at around 32 °C and 60% RH. 

The odorants of different dosages were prepared as mentioned in Section 2.4. An aliquot (10 μL) of the test sample was added onto a piece of filter paper, exposed to the air for 30 s to allow solvent evaporation, and then placed in one odor chamber of the olfactometer, while the same amount of hexane (solvent control) was applied in the other odor chamber. In each test, a single *Varroa* mite was introduced into the marked point of the olfactometer and allowed 6 min to choose between the two branches. The movement of a tested mite was immediately recorded for 6 min using an iPhone 6 s (Apple Inc., Cupertino, CA, USA). The branch that the mite walked into was first noted, and then the distance between the mite’s stop point and the odor chamber was measured. One mite might alternate its walking between branches many times, however, only its first choice was taken into account. A choice was scored positive when a mite entered the arm and either stayed in the odor chamber or walked within 3 cm of the odor chamber. If one mite remained in the stem and did not make a choice within 6 min, it was recorded as “no choice” and not included in the following data analysis. Individual mites were used only once, and at least 30 mites were tested for each treatment. The two odor chambers were randomly reversed to avoid any positional bias. The Y-tube and glass odor chambers were cleaned with ultrapure water, acetone and hexane and then baked in an oven at 80 ℃ for at least 60 min before each trial. 

### 2.6. Statistical Analyses

Since the assumptions of analysis of variance (ANOVA) were not met with the data for the amount of four esters and the ETG responses, the nonparametric Kruskal–Wallis test was performed to determine the significant differences, and Dunn’s test was used to carry out the post hoc pairwise comparisons. A Chi-square test was applied to analyze the behavioral data from Y-tube olfactometer bioassays. In all cases, a *p* value < 0.05 was taken to be significant. All statistical analyses were performed using GraphPad Prism software version 8.0 (GRAPH PAD Software Inc, San Diego, CA, USA).

## 3. Results

### 3.1. Identification and Comparison of Body Surface Volatiles of Four Kinds of Honey Bee Larvae

In total, there were 63 volatile compounds identified from the early fifth instar worker and drone larvae of the two honey bee species (Appendix A). Compared to *A. cerana*, many more compounds were detected from both worker (36 vs. 17) and drone (36 vs. 20) larvae of *A. mellifera*, suggesting that this new host expressed more complicated odorant stimuli. Additionally, while the majority (>90%) of total volatiles were not constantly presented in all four types of honey bee larvae, there were four aliphatic esters, ethyl myristate (EM), methyl palmitate (MP), ethyl palmitate (EP), and ethyl oleate (EO) shared by them (Figure 1). 

Further quantitative and comparative analyses were carried out on the levels of these four shared substances among worker and drone larvae of both *A. mellifera* and *A. cerana*. Generally, a significant difference was detected for all four chemicals (Figure 2. Kruskal–Wallis test. For EM: H = 21.39, df = 3, *p* < 0.0001; For MP: H = 32.29, df = 3, *p* < 0.0001; for EP: H = 37.00, df = 3, *p* < 0.0001; for EO: H = 15.49, df = 3, *p* = 0.0014). For intraspecific comparisons, we found that Ac worker larvae had equal amounts of EM, EP, and EO to Ac drone larvae, but significantly higher level of MP than Ac drone larvae (Dunn’s test. MP: *p* = 0.0032), while Am worker larvae displayed significantly lower levels of all four compounds than Am drone larvae (Dunn’s test. EM: *p* = 0.0028; MP: *p* = 0.0031; EP: *p* < 0.0001; EO: *p* = 0.0022). For interspecific comparisons, Ac worker larvae showed similar levels of EM to Am worker larvae, but significantly higher amount of MP, EP, and EO than Am worker larvae (Dunn’s test. MP: *p < 0.0001*; EP: *p* = 0.0002; EO: *p* = 0.0159), while Ac drone larvae had similar levels of MP, EP, and EO to Am drone larvae, but significantly lower amount of EM than Am drone larvae (Dunn’s test. EM: *p* = 0.0127).

### 3.2. Electrophysiological Responses of Varroa destructor to the Identified Aliphatic Esters

ETG recordings showed that *V. destructor* responded significantly to all the four aliphatic esters (Figure 3. Kruskal–Wallis test. For EM: H = 16.58, df = 5, *p* = 0.0054; For MP: H = 25.66, df = 5, *p* = 0.0001; for EP: H = 27.29, df = 5, *p* < 0.0001; for EO: H = 15.07, df = 5, *p* = 0.0101). Specifically, *Varroa* mites had responses to three dosages (0.1, 1, 10 µg) of EM (Dunn’s test. 0.1 µg: *p* = 0.0139; 1 µg: *p* = 0.0102; 10 µg: *p* = 0.0013), but no responses to the other two higher dosage (100, 1000 µg). For MP, *Varroa* mites responded to all dosages except 1 µg (Dunn’s test. 0.1 µg: *p* = 0.0013; 10 µg: *p* = 0.0201; 100 µg: *p* < 0.0001; 1000 µg: *p* = 0.0382). For EP, *Varroa* mites expressed remarkable responses to all the higher dosages (Dunn’s test. 1 µg: *p* = 0.0231; 10 µg: *p* = 0.0280; 100 µg: *p* < 0.0001; 1000 µg: *p* = 0.0005), but no response to the lowest dosage. Last, there were three dosages of EO resulting in noticeable responses in *Varroa* mites (Dunn’s test. 1 µg: *p* = 0.0038; 100 µg: *p* = 0.0154; 1000 µg: *p* = 0.0091). 

### 3.3. Olfactory Responses of Varroa destructor to the Identified Aliphatic Esters

We used Y-tube olfactometer assays to detect the olfactory responses of *V. destructor* to EM, MP, EP, and EO. Behavioral data showed that EM significantly attracted *V. destructor* at a medium dosage of 10 µg (Figure 4A. Chi-square test, χ^2^ = 9.135, df = 1, *p* = 0.0025), but repelled the parasites at an ultra-high dosage of 1000 µg (χ^2^ = 7.333, df = 1, *p* = 0.0068). On the other hand, *V. destructor* had no preference for MP regardless of tested dosages (Figure 4B. Chi-square test, *p* > 0.05 for all five treatment concentrations). In contrast to the results observed with EM and MP, EP significantly repelled *V. destructor* at dosages of 10 µg and 100 µg (Figure 4C. Chi-square test, 10 µg: χ^2^ = 4.500, df = 1, *p* = 0.0339; 100 µg: χ^2^ = 8.000, df = 1, *p* = 0.0047). Similarly, *V. destructor* was significantly repelled by EO from low to high dosages (Figure 4D. Chi-square test, 0.1 µg: χ^2^ = 16.00, df = 1, *p* < 0.0001; 10 µg: χ^2^ = 6.250, df = 1, *p* = 0.0124; 100 µg: χ^2^ = 4.267, df = 1, *p* = 0.0389; 1000 µg: χ^2^ = 17.07, df = 1, *p* < 0.0001).

## 4. Discussion

*Varroa destructor* originally parasitized the Eastern honey bee *Apis cerana*, and then made a host shift to the Western honey bee *A. mellifera* in East Asia in the first half of the twentieth century, and spread rapidly across the world via global trade [11,41]. Both worker and drone larvae of the old and new hosts can attract the parasite [34,35], which suggests that common stimuli may exist among different host larvae. It was expected that the active chemicals affecting *Varroa* host finding would be identified more efficiently by screening the shared compounds by honey bee larvae of distinct species and sexes. To this end, we first collected both worker and drone larvae of the two honey bee species from the same apiary, and then profiled the headspace volatiles individually. 

Once coming out from brood cells together with emerging bees, *Varroa* mites enter the phoretic stage and usually perform host-finding behavior as well, since they prefer nurse bees over both newly-emerged bees and foragers [42]. However, it seems more important for *Varroa* mites to find out the right larval hosts than the adult hosts, because successfully entering a larval cell is a key step for their reproduction and population development. In fact, much more efforts have been made to understand the details of this process. On the other hand, the body surface compounds of adult bees could potentially attract *Varroa* mites as well, and it would be interesting to see if the acting chemicals are kept the same between larval and adult bees. 

In this study, healthy and normal worker and drone larval samples were required for profiling the headspace volatiles. Generally, larval samples from *Varroa*-free colonies are much healthier than those from *Varroa*-infested colonies [9,10]. Therefore, no *Varroa* infestation was taken as a standard of healthy colonies, which were then selected for sample collection. It does not mean that individuals collected from *Varroa*-free colonies are less attractive than those from *Varroa*-infested colonies. In addition, it has been shown that the cuticular hydrocarbon profile of worker bees can be modified by the presence of *V. destructor* in brood cells [43]. As *A. cerana* has become resistant to *Varroa* mites after a long term of co-evolution [44], no or very few mites can be seen in *A. cerana* colonies, so all the *A. cerana* colonies were considered *Varroa*-free. 

There are two widely used methods for collecting chemicals from the body surface of honey bee larvae: hexane extraction [24,32,34,45], and headspace volatile enrichment [30,46,47,48]. SPME has evolved into a versatile sample preparation tool, supporting headspace volatile enrichment with high efficiency [49]. Here, we applied SPME technology to sample honey bee larval volatiles, which were analyzed by GC-MS. 

We identified more than 60 compounds from all larval headspace volatile samples. The composition of larval volatiles is quite different among the four types of honey bee larvae. In contrast, the majority of the cuticular hydrocarbons (CHCs) were shared by distinct larval types [34]. Despite this significant differentiation of larval volatiles, we found that four straight-chain fatty acid esters were constantly detected in all four larval types, the peaks of which were concentrated in the right end of total ion chromatograms (TICs). Among these, three esters, MP, EP, and EO are well known compounds serving as components of a brood pheromone [28,50,51,52,53], while EM is newly identified in this study. In addition to these esters, there were some other compounds shared by different larval types, whose function in *Varroa* host finding is still needing to be explored. 

It has long been established that the infestation rate of *Varroa* mites in *A. mellifera* drone cells is much higher than that in worker cells [25,26,54], which can be partially explained by the fact that drone larvae are more attractive than worker larvae [24]. Our results demonstrated that *A. mellifera* drone larvae contained a significantly higher amount of all four esters than worker larvae, which is consistent with previous studies [24,28] and supports the preferred invasion of drone cells by *Varroa*. 

Since *V. destructor* has no eyes and lives in a dark hive environment, chemical signals play an important role in keeping its tight link with the honey bee hosts [11,23]. The mite also lacks antennae, but its forelegs function as such [23]. Similarly, our ETG recording analyses also showed that *Varroa* forelegs could respond to all the identified fatty acid esters from larval headspace volatiles, offering a physiological basis for the next behavioral assays. Generally, the strength of the ETG responses goes up with the increase of testing concentrations within a range of dosages, which is determined by the innate characteristics of olfactory neurons [55]. We did see that different dosages of esters had different effects on *Varroa* behavior. The reasons for this phenomenon are complicated. For one compound, not all concentrations have a consistent behavioral effect, and one dosage can attract *Varroa*, but the higher dosages just repel the mites. This is partially related to the chemical itself. Y-tube olfactometer bioassay is a classical and widely-used method for investigating the behavioral responses of insects and mites to test odors [56,57]. Here, this method was also applied to determine how *Varroa* mites responded behaviorally to the four fatty acid esters. We tested the identified compounds individually. It is important to first figure out the role of the identified compounds in affecting *Varroa* host finding behavior when they were applied solely. It was expected that a chemical alone was able to highly attract/repel the mites. Then, some combinations among the compounds can be carried out in future to obtain higher efficiency in the attraction/repellence of *Varroa*. Moreover, it would be interesting to know if one identified compound is better than another one in affecting the host finding of *V. destructor*.

For the new compound EM, we found that it significantly attracted *Varroa* mites at medium dosage, while it repelled the mites at extra high dosage. In regard to MP, no response was displayed by *Varroa* mites, which is consistent with earlier findings [29,30,58], but is inconsistent with that of Calderone and Lin (2001) [59] and Le Conte et al. (1989) [24]. Surprisingly, both EP and EO showed a repellent effect on the mites, which also contradicts the findings from Calderone and Lin (2001) [59] and Le Conte et al. (1989) [24]. These differences could be due to different testing instruments, testing conditions, or *Varroa* mites of different stages or physiology. Therefore, it is necessary to have standards for *Varroa* olfactory-related behavioral bioassays, so that results from different laboratories can be compared. The behavioral assay indicated that the aliphatic esters were not always attractive to *Varroa* mites, and some of them showed repellent effect on the mites. It makes it difficult and sometimes confusing to predict the total effect of the esters on the mites. Actual honey bee hives are a complex odorant world, and these volatiles are always emitted and function together, which is not like the behavioral assay conducted in laboratory where the odors can be tested one by one. We speculate that when combined volatiles work together, one compound generates the main influence while others have no or weak effect on the mites. However, further verifications are needed when using different chemical combinations.

Chemical stimuli that affect *Varroa* host-finding behavior have drawn long-term and intensive studies around the world. However, considering the complexity and diversity of chemical cues, we believe that there will be more chemical cues identified in the future, especially with the rapid development of chemical analysis technologies. Despite more standardized behavioral bioassays conducted in the laboratories, further field tests for the effectiveness and applicability of the signaling compounds and their potential combinations should be encouraged to produce attractants/repellents for *Varroa* monitoring and control. Especially as a novel compound, EM provides excellent potential for developing *Varroa* attractants, which would be safe and useful for monitoring and controlling the *Varroa* population by efficiently attracting and arresting the parasites. 

## Figures and Tables

**Figure 1 insects-14-00024-f001:**
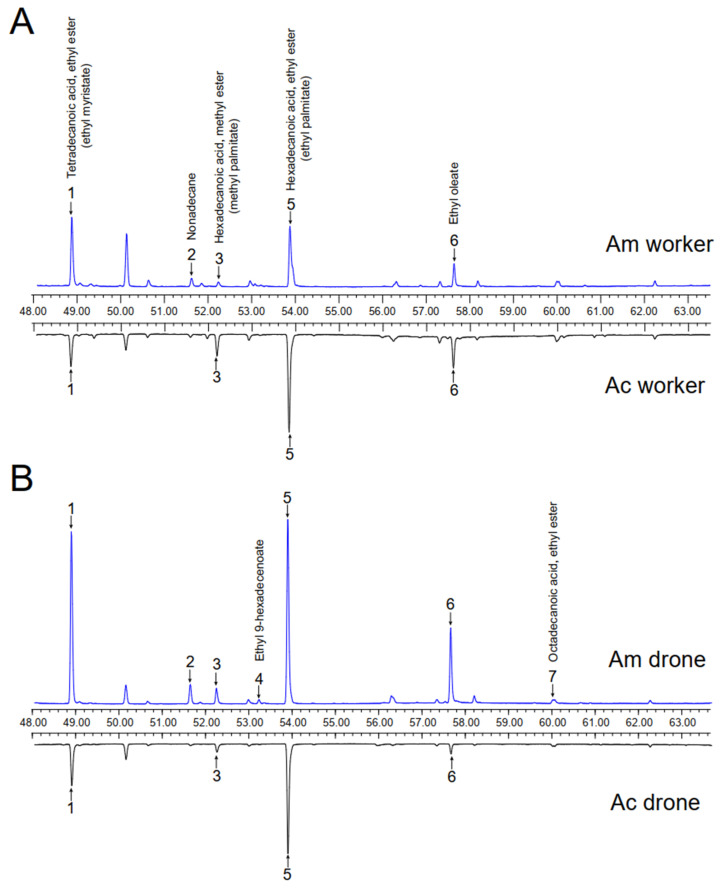
Total ion chromatograms (TIC) of headspace volatiles of *Apis mellifera* (Am) and *Apis cerana* (Ac) larvae. Only the right ends of the four TICs were shown and organized according to worker (**A**) and drone (**B**) larvae. The main compounds were: (1) tetradecanoic acid, ethyl ester (ethyl myristate); (2) nonadecane; (3) hexadecanoic acid, methyl ester (methyl palmitate); (4) ethyl 9-hexadecenoate; (5) hexadecanoic acid, ethyl ester (ethyl palmitate); (6) ethyl oleate; and (7) octadecanoic acid, ethyl ester.

**Figure 2 insects-14-00024-f002:**
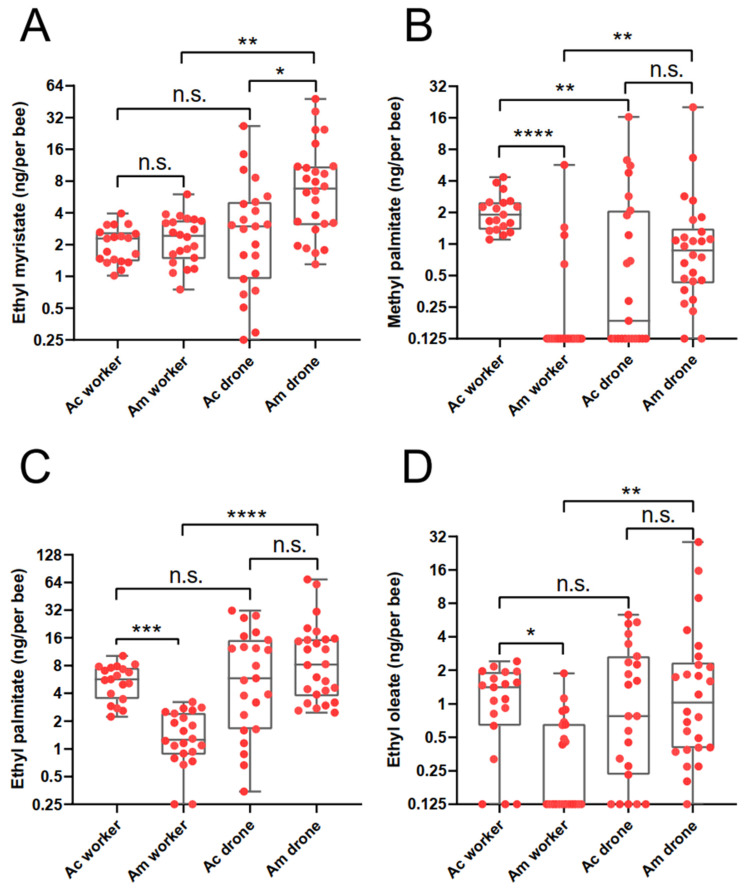
Comparison of the amount of four aliphatic esters ethyl myristate (**A**), methyl palmitate (**B**), ethyl palmitate (**C**), and ethyl oleate (**D**) among the worker and done larvae in both *Apis mellifera* (Am) and *Apis cerana* (Ac). Boxes include the 25th–75th percentile. Horizontal bars denote the median value. Whiskers indicate the values that are no further than 1.5 × IQR from either the upper or the lower hinge. Each dot represents one larval sample. The sample size (*n*) is 19, 23, 22, and 26 for Ac worker, Ac drone, Am worker, and Am drone, respectively. Statistical significance is determined by Kruskal–Wallis test followed by Dunn’s test for post hoc comparison. n.s. not significant, * *p* < 0.05, ** *p* < 0.01, *** *p* < 0.001, **** *p* < 0.0001.

**Figure 3 insects-14-00024-f003:**
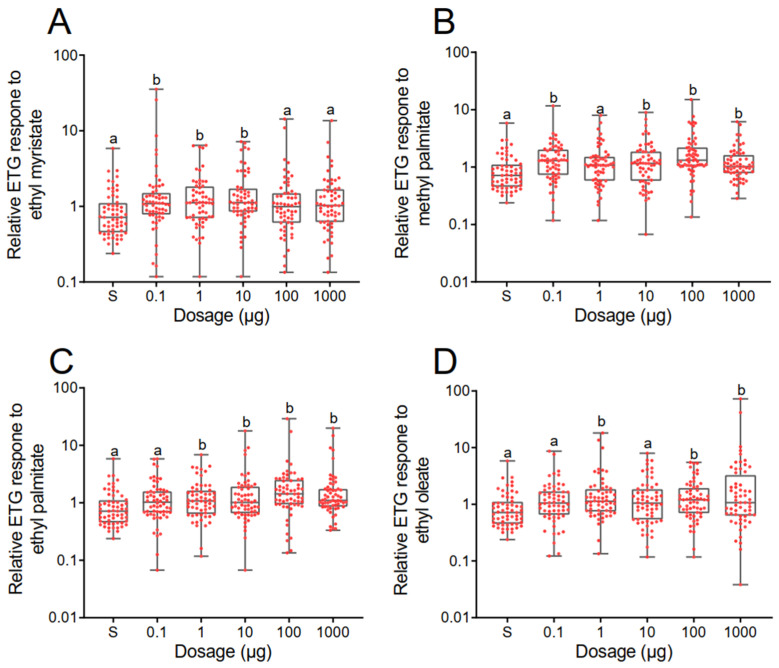
Electrophysiological responses of *Varroa destructor* to ethyl myristate (**A**), methyl palmitate (**B**), ethyl palmitate (**C**), and ethyl oleate (**D**). For each test chemical, five dosages (0.1, 1, 10, 100, 1000 µg) and solvent control (s) were included. All the responses to test compounds were calibrated to control (0.5). Each red dot represented one replicate (one mite) and 60 replicates (*n* = 60) were included for every single dosage and solvent control. Boxes include the 25th–75th percentile. Horizontal bars inside each box show the median value. Whiskers suggest values that are no further than 1.5 × IQR from either the upper or the lower hinge. Kruskal–Wallis tests were employed to determine the significant differences and Dunn’s tests were used for post hoc comparison. Every response to the test compounds was compared to solvent control, and a different lowercase letter was labeled to indicate significant differences (*p* < 0.05).

**Figure 4 insects-14-00024-f004:**
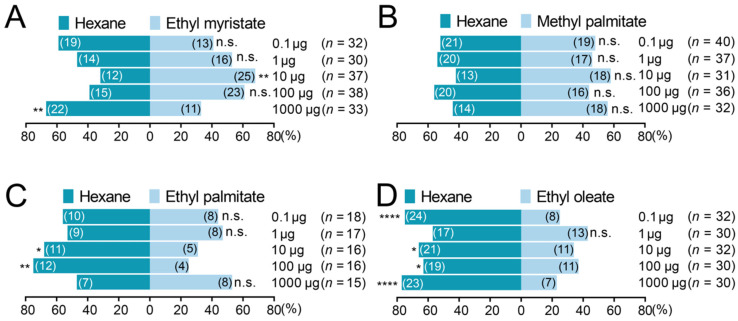
Olfactory responses of *Varroa destructor* to ethyl myristate (**A**), methyl palmitate (**B**), ethyl palmitate (**C**), and ethyl oleate (**D**) in Y-tube olfactometer assays. The numbers inside the bars represent the counts of *Varroa* mites selecting the corresponding arm data were analyzed using Chi-square test. * *p* < 0.05, ** *p* < 0.01, **** *p* < 0.0001, n.s. not significant.

## Data Availability

The data presented in this study are available on request from the corresponding author.

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
