# Peer review of "The Role of Honey Bee Derived Aliphatic Esters in the Host-Finding Behavior of Varroa destructor"

_insects, 2022, doi:10.3390/insects14010024_

Round 1

Reviewer 1 Report

In this manuscript, the authors study the role of honey bee derived aliphatic esters in the host-finding behavior of Varroa destructor, an ectoparasite of honey bees. The paper is well written, really easy to read, and will be an important contribution to the chemical ecology community and to everybody interested in parasitism or social behavior.

I have only a few comments that might be addressed to improve the manuscript before its publication in Insects.

-          The authors performed their experiments using honey bee’s larvae from 2 species to test the effects of different volatiles on the behavior and on the physiological responses of Varroa. Why did the authors only focused on larvae emitted volatiles? As mentioned L. 69-70, the Varroa has 2 different phases, with the 1st one (phoretic phase) in which the Varroa attach to adult bees and feed on them. They thus also need to find the adult workers, and could potentially detect workers emitted volatiles to find them. Please comment on that.

-          The authors mention that they used worker and drone larvae obtained from Varroa-free colonies. Why did they make this choice? Wouldn’t that mean that these individuals are less attractive than individuals from Varroa-infested colonies? Moreover, do the individuals from Varroa-free and Varroa-infested colonies emit the same volatile compounds? Finally, the authors did not classify the A. cerana colonies, why this difference? Please clarify these points.

-          The authors do not explain before the discussion why they record the physiological responses on the foreleg. The explanation L.323-324 should appear in the introduction.

-          What is recorded in the behavioral assay is not really clearly stated in the manuscript. The authors explain L.191-193 that “a choice was scored positive when a mite entered the arm and either stayed in the odor chamber or walked within 3cm of the odor chamber”. What is really recorded here? Do the authors mean the first choice of the mite? Please add details on the behavioral assays protocol. Moreover, the authors could have recorded the time spent in each arm during the 6min recording and show that they spend either more or less time than in the solvent chamber.

-          The results explain L.213-216 (and L.310) that there is 4 aliphatic esters shared by the 4 types of honey bee larvae. Are these volatiles the only ones shared by the 4 types of honey bee larvae? Or are there other volatiles in this case? Please clarify this point.

-          The authors claim L. 259 that EM, EP and EO induce a dosage-dependent response “to a certain extent”. Did the authors try to make more analysis on these results by fitting dose-response curves?

-          The authors do not really give a clear explanation of their results, it remains quite descriptive, and the link between the emitted volatiles and the behavior is lacking. They mention L.315-320 that it is known that drones are more attractive than workers for Varroa, an effect which could be explained by the fact that drones emit higher amount of all 4 esters than worker larvae. However, even if drones emit more EM which seems to have an attractive effect, 2 out of the 4 esters (EP and EO) seem to have a repellent effect. On average, drones would be more attractive but also more repellent for Varroa. Could the authors comment on this?

-          In the same line, how could the authors explain the different effects of the different concentrations, both on the ETG responses and on the behavior? Please comment in the manuscript.

Other minor points

-      L. 35: Change “utilized” by “used” please.

-      L.39: “honey bee cuticular volatiles”. The authors cannot be sure that the EM is part of the cuticle, as it could come from pheromonal glands. Please modify.

-   L. 56: Please change “factors” by “parasites”, otherwise the pesticides is certainly the main harmful factor for the Western honey bee.

-      L. 91: What do the authors mean by “arrested”?

-   At which temperature the ETG recordings were perform? Please add this detail.

Author Response

Response to Reviewer 1:

In this manuscript, the authors study the role of honey bee derived aliphatic esters in the host-finding behavior of Varroa destructor, an ectoparasite of honey bees. The paper is well written, really easy to read, and will be an important contribution to the chemical ecology community and to everybody interested in parasitism or social behavior. I have only a few comments that might be addressed to improve the manuscript before its publication in Insects.

    Thanks for the positive and encouraging evaluation on our work, and the comments are definitely helpful for improving the manuscript.

Major comments:

  1. The authors performed their experiments using honey bee’s larvae from 2 species to test the effects of different volatiles on the behavior and on the physiological responses of Varroa. Why did the authors only focused on larvae emitted volatiles? As mentioned L. 69-70, the Varroa has 2 different phases, with the 1st one (phoretic phase) in which the Varroa attach to adult bees and feed on them. They thus also need to find the adult workers, and could potentially detect workers emitted volatiles to find them. Please comment on that.

    Thanks for pointing out this good point. Once coming out from brood cells together with emerging bees, Varroa mites enter the phoretic stage and usually perform host-finding behavior as well, since they prefer nurse bees over both newly-emerged bees and foragers (Xie et al., 2016). However, it seems more important for Varroa mites to find out the right larval hosts than the adult hosts, because successfully entering a larval cell is a key step for their reproduction and population development. In fact, much more efforts have been made to understand the details of this process. On the other hand, we do believe that body surface compounds of adult bees could potentially attract Varroa mites, and it might be interesting to see if the acting chemicals are kept the same between larval and adult bees.

  1. The authors mention that they used worker and drone larvae obtained from Varroa-free colonies. Why did they make this choice? Wouldn’t that mean that these individuals are less attractive than individuals from Varroa-infested colonies? Moreover, do the individuals from Varroa-free and Varroa-infested colonies emit the same volatile compounds? Finally, the authors did not classify the A. cerana colonies, why this difference? Please clarify these points.

    Thanks. In this study, healthy and normal worker and drone larval samples were required for profiling the headspace volatiles. Generally, larval samples from Varroa-free colonies are much healthier than those from Varroa-infested colonies. Therefore, No Varroa infestation was taken as a standard of healthy colonies, which were then used for sample collection. It does not mean and no evidence shows that individuals collected from Varroa-free colonies are less attractive than those from Varroa-infested colonies. We find out that one previous study did show that the cuticular hydrocarbon profile of worker bees could be modified by the presence of Varroa destructor in brood cells (Salvy et al., 2001). Because A. cerana have become resistant to Varroa mites after a long term of co-evolution, no or very few mites can be seen in A. cerana colonies, so all the A. cerana colonies were considered Varroa-free.

  1. The authors do not explain before the discussion why they record the physiological responses on the foreleg. The explanation L.323-324 should appear in the introduction.

    Thanks for the helpful comments. Now we have explained why we recorded the electrophysiological responses on Varroa forelegs in the last paragraph of Introduction as such “The forelegs of V. destructor were used to record the electrophysiological responses as the pit organ located on the foreleg tarsi was found to sense the headspace volatiles of adult honey bees”. Please find this update in the revised manuscript.

  1. What is recorded in the behavioral assay is not really clearly stated in the manuscript. The authors explain L.191-193 that “a choice was scored positive when a mite entered the arm and either stayed in the odor chamber or walked within 3cm of the odor chamber”. What is really recorded here? Do the authors mean the first choice of the mite? Please add details on the behavioral assays protocol. Moreover, the authors could have recorded the time spent in each arm during the 6min recording and show that they spend either more or less time than in the solvent chamber.

    Thanks for pointing it out. In the behavioral assay, we recorded the whole 6 min movement of the test mite in each test by video. For the description in L191-193, we first noted which branch the mite walked in, and then measured the distance between the mite’s stop point after walking and the odor chamber. One mite may switch its walking many times, however, only its first choice was taken into account. All these details have been included in Section 2.5 in the revised manuscript. To compare the time that the mites spend in the odor chamber to that in the solvent chamber might be another alternative way to judge the behavioral selection of Varroa mites.

  1. The results explain L.213-216 (and L.310) that there is 4 aliphatic esters shared by the 4 types of honey bee larvae. Are these volatiles the only ones shared by the 4 types of honey bee larvae? Or are there other volatiles in this case? Please clarify this point.

    Thanks. There is another compound, Oxime-, methoxy-phenyl-_, shared by all four types of honey bee larvae (Table S1). But for aliphatic esters, only the mentioned four chemicals were found throughout the four types of hosts.

  1. The authors claim L.259 that EM, EP and EO induce a dosage-dependent response “to a certain extent”. Did the authors try to make more analysis on these results by fitting dose-response curves?

    Thanks for pointing it out. We have to say that we did not further analyze the data by fitting dose-response curves, so that we feel that it was not appropriate to make this claim. Finally, we delete it in the revised version.

  1. 7. The authors do not really give a clear explanation of their results, it remains quite descriptive, and the link between the emitted volatiles and the behavior is lacking. They mention L.315-320 that it is known that drones are more attractive than workers for Varroa, an effect which could be explained by the fact that drones emit higher amount of all 4 esters than worker larvae. However, even if drones emit more EM which seems to have an attractive effect, 2 out of the 4 esters (EP and EO) seem to have a repellent effect. On average, drones would be more attractive but also more repellent for Varroa. Could the authors comment on this?

    Thanks for these helpful comments. The behavioral assay indicated that the aliphatic esters were not always attractive to Varroa mites, and some of them showed repellent effect on the mites. It makes it difficult and sometimes confusing to predict the total effect of the esters on the mites. Actual honey bee hives are a complex odorant world, and these volatiles are always emitted and function together, which is not like the behavioral assay we conducted in lab where we can test the odor one by one. We speculate that when combined volatiles work together, one compound generates the main influence while others have no or weak effect on the mites. However, further verifications are needed using different chemical combinations.

  1. In the same line, how could the authors explain the different effects of the different concentrations, both on the ETG responses and on the behavior? Please comment in the manuscript.

    Thanks. Here, we first have to make it clear that in Figure 3 the ETG responses to different concentrations were compared to the responses to solution control, but not with each other. Generally, the strength of the ETG responses goes up with the increase of testing concentrations within a range of dosages, which is determined by the innate characteristics of olfactory neurons. We did see that different dosages of esters had different effects on Varroa behavior. The reason for this phenomenon is complicated. For one compound, not all concentrations have consistent behavioral effect, and one dosage can attract Varroa, but the higher dosages just repel the mites. This is partially related to the chemical itself.

Other minor points:

  1. L.35: Change “utilized” by “used” please.

    Thanks. Have been changed.

  1. L.39: “honey bee cuticular volatiles”. The authors cannot be sure that the EM is part of the cuticle, as it could come from pheromonal glands. Please modify.

    Thanks. We have changed it to “honey bee headspace volatiles”.

  1. L.56: Please change “factors” by “parasites”, otherwise the pesticides is certainly the main harmful factor for the Western honey bee.

    Thanks. Have been changed.

  1. L.91: What do the authors mean by “arrested”?

    Thanks. Here, “arrested” means mites were attracted to and then always stayed in the area with simple odd numbered C19-C29 n-alkanes.

  1. At which temperature the ETG recordings were performed? Please add this detail.

    Thanks. We performed the ETG recordings at room temperature. The information has been added to Section 2.4.

Reviewer 2 Report

The manuscript is well written and provides valuable information to the readers.

Need to add the GC-MS Table with RT, %, KI index (both literature and experiment )

Future Recommendation 

Author Response

Response to Reviewer 2:

The manuscript is well written and provides valuable information to the readers.

    Thanks for the positive evaluation on our work.

Major comments:

  1. Need to add the GC-MS Table with RT, %, KI index (both literature and experiment )

    Thanks for the helpful suggestion. The RT (retention time) is already presented in the second column. The percentage (%) of peak area has been used to calculate the absolute amount of each compound in every bee samples, which is used to show the chemical quantities. We feel sorry that we are not familiar with the KI index, and may not able to provide it with the table.

Future Recommendation:

  1. Start with importance and previous literature of inter specific and intra specific volatile commutations in insects. There are numerous reposts of VOC as a communications cues. 1. Volatile signals from guava plants prime defense signaling and increase jasmonate-dependent herbivore resistance in neighboring citrus plants.2. Interference mechanism of Sophora alopecuroides L. alkaloids extract on host finding and selection of the Asian citrus psyllid Diaphorina citri Kuwayama (Hemiptera: Psyllidae)

    Thanks for this helpful comment. We have talked about the importance of VOCs in the beginning of the fourth paragraph in Introduction. Please find the details in the revised manuscript.

  1. Mention peaks compounds in the graph for better understanding

    Thanks for the suggestion. Compound names have been shown in the graph as recommended in the revised version.

  1. What are the widely used method to control the Varroa destructor?

    Thanks. The widely used methods to control the Varroa mites include the chemical treatment with hard and soft acaricides. The hard acaricides are the organophosphate coumaphos, the pyrethroids tau-fluvalinate and flumethrin, as well as the formamidine amitraz. The soft acaricides contain the organic acids (formic acid and oxalic acid) and essential oils (thymol).

  1. Add future commendation

    Thanks. We have added one recommendation that we suggest that ethyl myristate can be used to generate a repellent in future. Please find the update in the revised manuscript.